# SCALE CONTINUITY IN GRAPH LEARNING: GOING BEYOND SPECTRAL METHODS

## ABSTRACT

Standard message passing graph neural networks (MPNNs) assign vastly different latent embeddings to graphs that describe the same underlying object at different resolution scales. As a result, message passing graph neural networks generically fail to generalize across resolution scales. Previous work showed that this issue can be overcome if instead of MPNNs certain special types of spectral graph neural networks are used. In this tiny-paper, we show that spectral methods are not necessary for achieving scale continuity. We demonstrate that if message passing networks are suitably modified, they can be rendered scale-continuous as well. By identifying the structural requirements for continuity, we derive a class of message passing architectures that provably preserve embeddings across resolution scales. Empirically, we show that these models match the cross-scale generalization performance of spectral approaches while retaining the flexibility and scalability of local message passing methods.

## 1 INTRODUCTION

A key theoretical pillar underpinning the widespread adoption of graph neural networks is the commonly held belief that graph neural networks (GNNs) define *stable* and *continuous* mappings: Small changes in the input graph are expected to induce correspondingly small changes in learned representations (Ruiz et al., 2020; Maskey et al., 2021; Roddenberry et al., 2022; Le & Jegelka, 2023). Recently this view has been challenged in the setting of stability under variations in graph resolution scale: In Koke et al. (2024; 2025) it was established that common graph neural networks based on message passing layers are not continuous under graph coarsening operations. Latent embeddings $F, \underline{F}$ generated by MPNNs for a graph $G$ and its coarsified version $\underline{G}$ are vastly different. The root cause behind this discontinuity across scales was identified to lie in a fundamental limitation of standard message passing neural networks. To overcome this limitation and build models that *are* able to generalize across scales, Koke et al. (2024; 2025) took a spectral approach and showed that if the layer-wise update rule of a node-feature matrix $X$ inside a *spectral* GNN is implemented as

$$X \mapsto \sum_{k=1}^{K} \psi_k(L) X W_k, \tag{1}$$

the resulting network will be scale continuous; assigning similar latent embeddings ($\underline{F} \approx F$) to graphs $G, \underline{G}$ describing the same object at different resolutions. Here $L$ is the natural ("un-normalized") graph Laplacian (c.f. also Appendix A.1), and the functions $\psi_k$ applied to $L$ have a specific form (arising as Laplace transforms; c.f. Section 3 below).

These works also showed (c.f. e.g. Koke et al. (2025, Appendix B.1)) that *standard* MPNNs *never* are scale continuous, generating vastly different latent embeddings $F, \underline{F}$ for similar graphs $G, \underline{G}$ describing the same object at different resolutions. This might lead one to assume that only spectral graph networks (with layer-wise update as defined in (1)) can be scale continuous. Here we however show that this is not the case: **Also MPNNs may be rendered scale continuous**, if message passing is implemented appropriately. Concretely, we show that MPNNs with layer-wise update

$$h_u^{(\ell+1)} = \sum_{v \in G} m_{uv}, \qquad m_{uv} = [\psi(L)]_{vu} \cdot \phi(h_u^{(\ell)}, h_v^{(\ell)}), \qquad h_0 = \psi(L)X, \tag{2}$$

*are* in fact scale continuous. Here $h_u^{(\ell)}$ is the latent representation of node $u$ in layer $\ell$, $\phi(\cdot, \cdot)$ is a learnable message function and $\psi(L)$ is a specific (type of; c.f. Section 3) function applied to $L$.

## 2   RECAP: DISCONTINUITY OF STANDARD MPNNs ACROSS SCALES

To demonstrate that conventional GNN architectures struggle to coherently handle data across different resolutions, we adopt the experimental setting of (Koke et al., 2024; 2025) and make use of the QM7 benchmark dataset (Rupp et al., 2012). This dataset consists of small organic molecules composed of hydrogen and heavier elements, with the task of predicting their atomization energies. Each molecule is encoded as a weighted graph whose adjacency matrix is defined by $A_{ij} = Z_i Z_j |\vec{x}_i - \vec{x}_j|^{-1}$, representing the Coulomb interaction energy between atoms $i$ and $j$.

From a physical perspective, describing a molecule at the level of interacting atoms corresponds to a specific choice of resolution scale, where interactions of individual protons and neutrons inside individual atoms are discarded. To test the sale continuity of GNNs we also consider a version of QM7 where we further lower the resolution scale: Here we aggregate each heavy atomic core together with its surrounding (single-proton) hydrogen atoms into super-nodes.

To showcase the failure of GNNs to consistently incorporate multiple scales, we confront models during inference with a version of QM7 on a scale different from the one they were trained on. As summarized in Table 1, mean absolute errors rise sharply when models are transferred from a same-resolution- to a cross-resolution setting. No architectures —including pooling- or multiscale variants (SAG-M through PushNet)— are able to reliably process multiple scales. This can be traced back to the latent embeddings $\{F\}$ and $\{\underline{F}\}$ that are being generated for original- $\{G\}$ and coarsified graphs $\{\underline{G}\}$: For models of Table 1 on average $10 \lesssim \|F - \underline{F}\| \lesssim 10^3$ (c.f. also Table 2 below): Latent representations of graphs describing the same object at different resolutions differ significntly.

As argued in (Koke et al., 2024; 2025) the underlying reason for this difference in latent embeddings is that standard GNNs are not continuous. To see this, we interpolate between fine and coarse resolution: Original graphs $\{G\}$ of QM7 are modified ($\{G_\omega\}$) by moving hydrogen atoms towards their corresponding heavy atom by a factor of $\omega \geq 1$ (i.e. dist$_\text{new}$ = dist$_\text{equilib.}/\omega$). For $\omega \to \infty$, they arrive at the respective heavy atom ($\{\underline{G}\}$). In Fig. 2 we compare the latent distance between the coarse embeddings $\underline{F}$ and the embeddings for the intermediate graph $F_\omega$. Embeddings $F_\omega$ do *not* converge to the coarse embeddings $\underline{F}$. Since the convergence of graph-sequence $G_\omega$ to the limit graph $\underline{G}$ is not turned into a convergence of latent embeddings $F_\omega \nrightarrow \underline{F}$ we conclude: GNNs are not continuous in this setting. This discontinuity explains why GNNs can map similar graphs (describing the same object at different resolutions) to different latent representations.

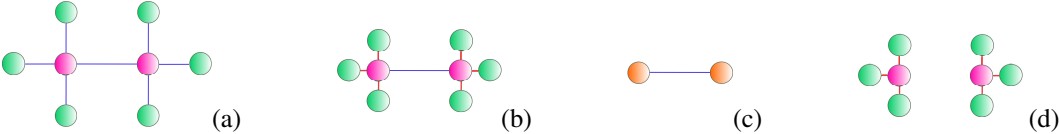

Figure 1: (a) $G$ of QM7    (b) Modified $G_\omega$    (c) Coarsified $\underline{G}$    (d) Effective propagation in GCN

Table 1: QM7 regression. Mean Absolute Error

| Resolution | MAE ($\downarrow$) on QM7 [kcal/mol] | | | |
|---|---|---|---|---|
| Training: | **Fine** | | **Coarse** | |
| Inference: | **Coarse** | Fine | Coarse | **Fine** |
| GCN | **136.7**$\pm_{6.6}$ | (63.6$\pm_{1.3}$) | (63.6$\pm_{1.3}$) | **138.1**$\pm_{2.4}$ |
| GATv2 | **423.5**$\pm_{337.1}$ | (67.4$\pm_{8.2}$) | (59.7$\pm_{2.7}$) | **257.4**$\pm_{139.1}$ |
| ChebNet | **447.8**$\pm_{6.0}$ | (66.7$\pm_{1.4}$) | (71.5$\pm_{2.1}$) | **158.7**$\pm_{57.4}$ |
| GIN | **658.4**$\pm_{85.8}$ | (17.$\pm_{2.8}$) | (38.3$\pm_{21.6}$) | **1835.4**$\pm_{925.8}$ |
| SAG | **589.7**$\pm_{44.9}$ | (68.2$\pm_{2.6}$) | (107.9$\pm_{1.1}$) | **283.6**$\pm_{39.3}$ |
| SAG-M | **194.4**$\pm_{29.9}$ | (66.6$\pm_{1.9}$) | (77.8$\pm_{6.0}$) | **219.5**$\pm_{11.7}$ |
| UFGNet | **131.5**$\pm_{6.1}$ | (62.4$\pm_{0.7}$) | (69.4$\pm_{0.7}$) | **148.1**$\pm_{6.3}$ |
| Lanczos | **938.4**$\pm_{2.5}$ | (9.9$\pm_{2.5}$) | (88.2$\pm_{2.7}$) | **658.6**$\pm_{199.2}$ |

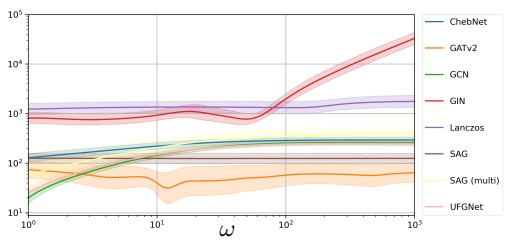

Figure 2: Latent distance $\|F_\omega - \underline{F}\|$.

To understand this discontinuity, one may exemplarily investigate GCN (Kipf & Welling, 2017). There the layer-wise update acts as $X \mapsto \hat{A}XW$, with the feature matrix $X \in \mathbb{R}^{N \times F}$ ($N$ nodes; latent dimension $F$), the weight matrix $W \in \mathbb{R}^{F \times F}$ and the *renormalized* adjacency matrix $\hat{A} \in \mathbb{R}^{N \times N}$. As hydrogen atoms move closer to heavy atoms, the entries $\hat{A}_\text{heavy,heavy}$ in $\hat{A}_{ij} \sim A_{ij}/\sqrt{d_i d_j}$ tend to zero (degrees $d_\text{heavy}$ tend to infinity). This disrupts information-flow over the graph (Fig 1 (d)).

# 3    GLOBAL LAPLACE PROPAGATION IN GNNS

To avoid disconnected limit propagation as in Fig. 1 (d), GCN's propagation scheme is then modified:

## 3.1    SPECTRAL NETWORKS

To connect the information flows over $G_\omega$ and $\underline{G}$ one may note that features in $G_\omega$ should equalize faster between nodes connected by large edge weights. When such a large weight tends to infinity, features between strongly connected nodes are then equalized immediately, so that entire strongly connected clusters exactly behave as the single nodes in $\underline{G}$. This is exactly the behavior that heat dissipating over a graph exhibits. Hence we may make use of the the heat diffusion equation $dX(t)/dt = -L \cdot X(t)$ (with Graph Laplacian $L$ and time $t$) and the structure $X(t) = e^{-Lt} \cdot X(0)$ of its solutions, when designing scale continuous graph networks: Let $\hat{\psi}$ be a bounded (generalized) function defined on $[0, \infty)$. A **Global Laplacian Propagation Matrix** $\psi(\mathrm{L})$ is any matrix arising as $\psi(L) := \int_0^\infty e^{-tL}\hat{\psi}(t)dt$. Thus $\psi(L)$ represent a weighted sum of diffusion flows that have progressed to various times. Specifically, choosing the Dirac distribution $\hat{\psi}_{\delta_{t_k}}(t) := \delta(t - t_k)$ as the weighting function $\hat{\psi}_k$, we obtain **exponential** matrices $\psi_k(L) = \int_0^\infty \delta(t - t_k)e^{-tL}dt = e^{-t_k L}$ and $\hat{\psi}_k := (-t)^{k-1}e^{-\lambda t}$ to get powers of **resolvents** $\psi_k(L) = [(zId + L)^{-1}]^k$.

In (Koke et al., 2024; 2025) it was then proved that if the update rule $X \mapsto \hat{A}XW$ of GCN is replaced with the update $X \mapsto \psi(L)XW$ (with $\psi(L)$ a Laplacian propagation matrix), the resulting networks are scale continuous (e.g. $\|F_\omega - F\| \to 0$ in the setting of Fig. 2). Furthermore, it was noted that GCN may be thought of as the simplest spectral network, which allowed to extend scale continuity results to more general spectral networks, with layer-wise update rule $X \mapsto \sum_{k=1}^K \psi_k(L)XW_k$.

## 3.2    MESSAGE PASSING NETWORKS:

Here we note that GCN is not only the simplest spectral GNN, but also the simplest *message-passing* based GNN. Hence we aim to extend previous results on how to turn spectral GNNs scale continuous also to the realm of message passing networks.

To determine which modifications exactly are needed in order to turn *message passing* based networks scale continuous, we gain intuition from the the binary edge setting (adjaceney entries $A_{uv} \in \{0, 1\}$): For networks with sum aggregation, the layer-wise update of a MPNN may then be written as $h_u^{(\ell+1)} = \sum_{v \in G} m_{uv}$, with message function $m_{uv} = A_{uv} \cdot \phi(h_v^{(k)}, h_u^{(k)})$. As we saw in Section 2, such adjacency-focused propagation schemes result in networks that are not continuous across scales. We hence make the replacement $A \mapsto \psi(L)$ in our propagation scheme:

$$h_u^{(\ell+1)} = \sum_{v \in G} m_{vu}, \qquad m_{uv} = [\psi(L)]_{uv} \cdot \phi(h_u^{(\ell)}, h_v^{(\ell)}), \qquad h_0 = \Psi(L)X, \qquad (3)$$

For the choice $\phi(h_u^{(\ell)}, h_v^{(\ell)}) = W \cdot h_v^{(\ell)}$ this reduces to the modified GCN version of Section 3.1.

**Scalability:**   At first glance, propagation along Laplace-transform matrices $\psi(L)$ may appear difficult to scale, since matrices such as $\psi(L) = e^{-tL}$ are typically dense. A naïve implementation would thus require sending $N^2$ messages over an $N$-node graph. This does however not pose a practical bottleneck: most entries are vanishingly small (Bauer et al., 2017). Hence, propagation remains effectively sparse, and negligible edges can be safely discarded in practice.

**Theoretical Guarentees:**   Modifying message-passing based graph neural networks exactly as outlined in (3) then indeed yields scale continuous networks, as we show next. Our main Theorem (proved in Appendix C.2) establishes scale-continuity of the modified MPNNs:

**Theorem 3.1.** *In the setting of Figure 2 let $F_\omega$ and $\underline{F}$ be the latent embeddings generated for graphs $G_\omega$ and $\underline{G}$ by a modified message passing network. For such latent embeddings we have $\|F_\omega - \underline{F}\| \to 0$ as $\omega$ increases.*

Hence the modified MPNNs of (3) are indeed continuous in the setting of Section 2 and thus assign similar latent embeddings $F \approx \underline{F}$ to graphs $G, \underline{G}$ describing the same object at different resolutions.

## 4 NUMERICAL INVESTIGATION OF SCALE CONTINUITY FOR MPNNS

**Continuity across scales:** Theorem 3.1 implies that MPNNs employing Laplace-transform based propagation schemes as in (3) are continuous as maps from the space of graphs into their latent spaces. To numerically verify this, we repeat the experiment of Figure 2 for models belonging to this category (using resolvent and exponential matrices; cf. Section 3 in a spectral and a spatial setting). As is evident from Fig. 3, latent embeddings generated by models employing Laplace transform propagation **including modified MPNN based methods** *do* indeed converge ($\|F_\omega - \underline{F}\| \to 0$). Hence for the message passing models constructed in Section 3.2, we have indeed verified the desired continuity across scales.

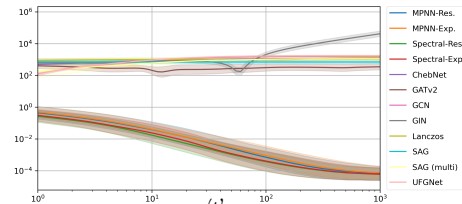

Figure 3: Latent distance $\|F_\omega - \underline{F}\|$. Latent embeddings generated by Laplace transform based GNNs converge; others do not.

**Resulting Generalization Ability:** In Section 2 we had identified lack of continuity as the obstruction to generalizing across scales. As verified above, MPNNs based on Laplace-transform propagation *are* continuous. Hence we expect them to map similar graphs to similar latent embeddings. To verify this, we here repeat the generalization experiment of Table 1 in Section 2.

Table 2: Embedding difference $\|F - \underline{F}\|$ across resolution scales averaged over 5 runs (mean±std). Lower is better ($\downarrow$).

| Training → Inference: | Fine → Coarse | Coarse → Fine |
|---|---|---|
| GCN | $20.3 \pm 1.6$ | $18.0 \pm 0.6$ |
| GATv2 | $76.8 \pm 52.4$ | $42.5 \pm 24.3$ |
| ChebNet | $123.0 \pm 3.8$ | $158.5 \pm 57.40$ |
| GIN | $881.2 \pm 268.7$ | $2875.8 \pm 1527.7$ |
| SAG | $113.5 \pm 10.3$ | $43.3 \pm 5.4$ |
| SAG-M | $75.0 \pm 15.1$ | $61.7 \pm 4.6$ |
| UFGNet | $24.5 \pm 0.3$ | $26.8 \pm 2.1$ |
| Lanczos | $1286.1 \pm 54.4$ | $728.4 \pm 230.0$ |
| Spectral$_{\text{exp.}}$ | $0.3 \pm 0.1$ | $0.3 \pm 0.1$ |
| Spectral$_{\text{res.}}$ | $0.3 \pm 0.1$ | $0.3 \pm 0.3$ |
| **MPNN$_{\text{exp.}}$** | $0.2 \pm 0.1$ | $0.2 \pm 0.1$ |
| **MPNN$_{\text{res.}}$** | $0.2 \pm 0.1$ | $0.2 \pm 0.2$ |

Table 3: QM7 regression. Mean Absolute Error (MAE $\downarrow$) in kcal/mol for training and inference at different resolutions scales.

| Resolution | MAE ($\downarrow$) on QM7 [kcal/mol] | | | |
|---|---|---|---|---|
| Training: | Fine | | Coarse | |
| Inference: | **Coarse** | Fine | Coarse | **Fine** |
| GCN | $\mathbf{136.7}_{\pm 6.6}$ | $(63.6_{\pm 1.3})$ | $(63.6_{\pm 1.3})$ | $\mathbf{138.1}_{\pm 2.4}$ |
| GATv2 | $\mathbf{423.5}_{\pm 337.1}$ | $(67.4_{\pm 8.2})$ | $(59.7_{\pm 2.7})$ | $\mathbf{257.4}_{\pm 139.1}$ |
| ChebNet | $\mathbf{447.8}_{\pm 6.0}$ | $(66.7_{\pm 1.4})$ | $(71.5_{\pm 2.1})$ | $\mathbf{158.7}_{\pm 57.4}$ |
| GIN | $\mathbf{658.4}_{\pm 85.8}$ | $(17._{\pm 2.8})$ | $(38.3_{\pm 21.6})$ | $\mathbf{1835.4}_{\pm 925.8}$ |
| SAG | $\mathbf{589.7}_{\pm 44.9}$ | $(68.2_{\pm 2.6})$ | $(107.9_{\pm 1.1})$ | $\mathbf{283.6}_{\pm 39.3}$ |
| SAG-M | $\mathbf{194.4}_{\pm 29.9}$ | $(66.6_{\pm 1.9})$ | $(77.8_{\pm 6.0})$ | $\mathbf{219.5}_{\pm 11.7}$ |
| UFGNet | $\mathbf{131.5}_{\pm 6.1}$ | $(62.4_{\pm 0.7})$ | $(69.4_{\pm 0.7})$ | $\mathbf{148.1}_{\pm 6.3}$ |
| Lanczos | $\mathbf{938.4}_{\pm 2.5}$ | $(9.9_{\pm 2.5})$ | $(88.2_{\pm 2.7})$ | $\mathbf{658.6}_{\pm 199.2}$ |
| Spectral$_{\text{exp.}}$ | $\mathbf{15.9}_{\pm 1.1}$ | $(15.9_{\pm 1.1})$ | $(16.0_{\pm 1.5})$ | $\mathbf{16.0}_{\pm 1.5}$ |
| Spectral$_{\text{res.}}$ | $\mathbf{17.2}_{\pm 3.1}$ | $(17.2_{\pm 3.1})$ | $(15.8_{\pm 1.4})$ | $\mathbf{15.8}_{\pm 1.4}$ |
| **MPNN$_{\text{exp.}}$** | $\mathbf{17.4}_{\pm 3.4}$ | $(17.4_{\pm 3.4})$ | $(17.9_{\pm 5.2})$ | $\mathbf{17.9}_{\pm 5.2}$ |
| **MPNN$_{\text{res.}}$** | $\mathbf{18.0}_{\pm 4.3}$ | $(18.0_{\pm 4.3})$ | $(17.6_{\pm 4.9})$ | $\mathbf{17.6}_{\pm 4.9}$ |

Comparing with standard GNNs in Table 2 we see that in cross-resolution settings the difference $\|F - \underline{F}\|$ of latent embeddings generated by methods employing global Laplacian propagation schemes are *lower* than those of standard graph learning methods by factors of order $10^2$ to $10^4$. Furthermore we note that indeed not only those of spectral methods, but **also embeddings generated MPNNs based on Laplace transform propagation are stable** ($\|F - \underline{F}\| \approx 0$).

The small-to-negligible variations in latent representations then translate into small-to-negligible variations in prediction performance: As we infer from Table 3 below, MAEs generated by Laplace-Transform based GNNs in the cross resolution setting are essentially the same as those corresponding to same-resolution settings. GNNs based on Laplace Transform propagation schemes are able to *generalize* across scales. This is indeed not limited to spectral methods: **Also modified MPNN-based methods are able to generalize across scales.**

## 5 DISCUSSION

We showed that scale continuity is not limited to spectral graph neural networks but can also be achieved with message passing models. By replacing adjacency-based propagation with Laplace-transform based information flow, MPNNs can be rendered provably continuous across graph resolutions. This unifies spectral and spatial approaches under a common principle: continuity depends on the propagation operator rather than on the network family. Empirically, the new resulting message passing based models match the cross-scale generalization ability of spectral methods.

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

## A GRAPHS AND GRAPH LEARNING

We first briefly review graphs and graph neural networks:

### A.1 GRAPHS AND THEIR DESCRIPTION

We consider finite, weighted graphs $G = (V, E)$ equipped with a *node weight matrix* $M = \text{diag}(\mu_v)_{v \in V}$. Each weight $\mu_v > 0$ reflects node-wise information. Examples are physical mass or charge of atom $v$ in a molecular graph, number of users represented if $v$ is a node in a social network, or a volume element if $v$ is part of a point cloud. Often, all node masses are set to $\mu_v = 1$ so that $M = I$. Edges $(u, v) \in E$ of a graph carry symmetric non-negative (potentially binary) weights $A_{uv}$ which we collect in the adjacency matrix $A$. The corresponding degree matrix is $D = \text{diag}(d_v)$ with $d_v = \sum_u A_{uv}$.

The natural self-adjoint Laplacian associated with a node-weighted graph is $L = M^{-1}(D - A)$. It is self-adjoint with respect to the $M$-weighted inner product $\langle f, g \rangle_M = f^\top M g$. When the node weights encode local volumes or sampling densities it may be thought of as the discrete counterpart of the Laplacian $L = -\Delta$ on a continuum domain (Chung, 1997). While the natural Laplacian is central to our work, more commonly used in machine learning applications is instead the normalized Laplacian $\mathcal{L} = I - D^{-1/2} A D^{-1/2}$. It is self-adjoint in the standard Euclidean inner product and its spectrum is contained in the interval $[0, 2]$.

### A.2 GRAPH NEURAL NETWORKS

Graph neural networks are machine learning methods adapted to handle graph structured data. Corresponding network architectures broadly fall into two categories:

**Message Passing Networks:** Message Passing Neural Networks (MPNNs) Gilmer et al. (2017); Bronstein et al. (2021) are arguably the most widely used class of GNNs. At each layer $k$, node features $h_v^{(k)}$ are updated by aggregating information from neighbouring nodes:

$$m_{vu}^{(k)} = \phi(h_v^{(k)}, h_u^{(k)}, A_{uv}) \tag{4}$$

$$h_v^{(k+1)} = \text{AGG}_{u \in \mathcal{N}(v)} m_{vu}. \tag{5}$$

Here $A_{uv}$ denotes edge weights. Architectutes differ in which message- and aggregation functions are being used (Xu et al., 2019; Velickovic et al., 2018; Kipf & Welling, 2017; Hamilton et al., 2017), with typcial choices for the neighborhood aggregation function AGG being sum, avg, or (seldom) max (Fey et al., 2025).

**Spectral Networks:** Spectral graph concolutional layers (Bruna et al., 2014; Defferrard et al., 2016) apply learnable filters $g_\theta(\mathcal{L})$ to node-wise information via matrix-multiplication. The filters themselves are composed of learnable scalar functions $g_\theta(\cdot)$ which are applied to the normalized graph Laplacian $\mathcal{L}$. In practice, the filter $g_\theta$ is not implemented via eigendecomposition but approximated via polynomials (He et al., 2022; 2021; Koke & Cremers, 2024; Levie et al., 2019b). The prototypical example is ChebNet, where filters are parametrized as $g_\theta(\lambda) \approx \sum_{\ell=0}^{K} \theta_\ell T_\ell(\lambda)$, with $T_\ell$ the $\ell^{\text{th}}$-Chebyshev polynomial. With node feature matrix $X$ and learnable weight matrices $\{W_\ell\}_\ell$, a spectral layer is then efficiently implemented as

$$X^{(k+1)} = \sum_{\ell=0}^{K} T_\ell(\mathcal{L}) X^{(k)} W_\ell. \tag{6}$$

# B   STABILITY OF LATENT REPRESENTATIONS GENERATED BY LAPLACE-TRANSFORM BASED GNNS FOR GRAPHS ON THE SAME NODE SET

## B.1   NODE LEVEL STABILITY

In this Section, we are concerned with two graphs $G_1$ and $G_2$ that share a common node set. We assume that their heat kernels are similar in the sense of

$$\|e^{-tL_1} - e^{-tL_2}\| \approx 0. \tag{7}$$

We are then interested in bounding the variation in latent representations generated by a GNN $\Phi_{\mathscr{W},\mathscr{B},\Psi}$ with weights $\mathscr{W}$, (potential) biases, and Laplace transform propagation matrices $\Psi$.

### B.1.1   SPECTRAL LAPLACE-TRANSFORM METHODS

We first consider spectral Laplace Transform based spectral GNNs:

**Theorem B.1.** Let $\Xi_{\mathscr{W},\mathscr{B},\Psi}$ be a $K$-layer deep spectral graph convolutional architecture. Assume in each layer $1 \le \ell \le K$ that $\sum_i \|W_i^\ell\| \le W$ and $\|B^\ell\| \le B$ with weight- and bias-matrices $\{W^\ell, B^\ell\}$. Assume the non-linearity $\rho$ is 1-Lipschitz. Choose a constant $C$ such that $C \ge \|\Psi_i(L_1)\|, \|\Psi_i(L_2)\|$ for all Laplacian propagation matrices $\Psi_i(L_1), \Psi_i(L_2)$. W.l.o.g. assume $CW > 1$. With this, we have with $\delta = \max_{i \in I}\{\|\Psi_i(L_1) - \Psi_i(L_2)\|\}$ and input node-feature matrix $X$, that

$$\|\Xi_{\mathscr{W},\mathscr{B},\Psi}(L, X) - \Xi_{\mathscr{W},\mathscr{B},\Psi}(\widetilde{L}, X)\| \le \left[ K \cdot C^K W^{K-1} \cdot \left( \|X\| + \frac{1}{CW - 1}B \right) \right] \cdot \delta. \tag{8}$$

*Proof.* For simplicity in notation, let us denote the hidden representations in the network corresponding to $L_2$ by $X_i^\ell$. With this, we note:

$$\|X_1^K - X_2^K\| \le \sum_{i \in I} \|\psi_i(L_1) - \psi_i(L_2)\| \cdot \|X_1^{K-1}\| \cdot \|W_i^K\| + \sum_{i \in I} \|\psi_i(L_2)\| \cdot \|X_2^{K-1} - X_1^{K-1}\| \cdot \|W_i^K\|$$
$$\tag{9}$$

$$\le \delta W\|X_1^{K-1}\| + CW\|X_1^{K-1} - X_2^{K-1}\| \tag{10}$$

$$\le \delta W\|X_1^{K-1}\| + CW\delta\|X_1^{K-2}\| + (CW)^2\|X_2^{K-1} - X_1^{K-1}\| \tag{11}$$

$$\le \frac{\delta}{C} \cdot \left( \sum_{\ell=1}^{K} (CW)^\ell \|X_1^{K-\ell}\| \right) \tag{12}$$

$$= \frac{\delta}{C} \cdot \left( \sum_{j=0}^{K-1} (CW)^{K-j} \|X_1^j\| \right) \tag{13}$$

Hence we need to bound the quantity $\|X_1^j\|$ in terms of $C, W, B$ and $X$.

We have

$$\|X_1^j\| \le \sum_i \|\psi_i(L_1)\| \cdot \|X_1^{j-1}\| \cdot \|W_i^j| + \|B^J\| \tag{14}$$

$$\le CW\|X_1^{j-1}\| + B \tag{15}$$

$$\le (CW)^2\|X_1^{j-2}\| + CWB + B \tag{16}$$

$$\le B \left( \sum_{k=0}^{j-1} (CW)^k \right) + (CW)^j\|X\| \tag{17}$$

$$= \begin{cases} B\frac{(CW)^j - 1}{CW - 1} + (CW)^j\|X_1\| & ; CW \ne 1 \\ jB + \|X\| & ; CW = 1 \end{cases}. \tag{18}$$

For the case $CW = 1$, we thus find

$$\|X_1^K - X_2^K\| \leq \frac{\delta}{C} \cdot \left( \sum_{j=0}^{K-1} (jB + \|X\|) \right) \tag{19}$$

$$= \frac{\delta}{C} \cdot \left( K\|X\| + B\frac{K(K-1)}{2} \right). \tag{20}$$

For the case $CW \neq 1$, we find

$$\|X_1^K - X_2^K\| \leq \frac{\delta}{C} \cdot \left( \sum_{j=0}^{K-1} (CW)^{K-j} \left[ B\frac{(CW)^j - 1}{CW - 1} + (CW)^j\|X\| \right] \right) \tag{21}$$

$$\tag{22}$$

For $CW > 1$, we may further estimate this as

$$\|X_1^K - X_2^K\| \leq \frac{\delta}{C} \cdot \left( \sum_{j=0}^{K-1} (CW)^{K-j} \left[ B\frac{(CW)^j - 1}{CW - 1} + (CW)^j\|X\| \right] \right) \tag{23}$$

$$\leq \delta \cdot \frac{K(CW)^K}{C} \left[ \frac{B}{CW - 1} + \|X\| \right]. \tag{24}$$

This proves the claim.      □

Next we note that we can estimate $\delta$ in terms of $\max_i\{\int_0^\infty \hat{\psi}_i(t)\|e^{-tL_1} - e^{tL_2}\|dt\}$.

**Lemma B.1.** *We have*

$$\delta \leq \max_i \left\{ \int_0^\infty \hat{\psi}_i(t)\|e^{-tL_1} - e^{tL_2}\|dt \right\}. \tag{25}$$

*Proof.* We note that by definition, we have

$$\psi_i(L) := \int_0^\infty e^{-tL}\hat{\psi}_i(t)dt. \tag{26}$$

From this we may infer

$$\|\psi_i(L_1) - \psi_i(L_2)\| = \left\| \int_0^\infty \hat{\psi}_i(t) \left[ e^{-tL_1} - e^{-tL_2} \right] dt \right\| \leq \int_0^\infty \hat{\psi}_i(t) \left\| e^{-tL_1} - e^{-tL_2} \right\| dt. \tag{27}$$

Taking the maximum over $i$ indexing all propagation matrices. yields the claim.

     □

Combining Lemma B.1 with Theorem B.1 then establishes a bound on node-wise latent embeddings in terms of the heat kernel distance $\left\| e^{-tL_1} - e^{-tL_2} \right\|$ for spectral Laplace transform based methods.

### B.1.2   MESSAGE PASSING BASED LAPLACE TRANSFORM METHODS:

We next establish similar results for Message passing networks. To this end, we first introduce some notation. We continue to denote the node-feature matrix by $X$. We have as layer-wise update the implemented the scheme

$$m_{vu}^{(\ell)} = [\psi(L)]_{vu} \cdot \phi(h_v^{(k)}, h_u^{(k)}) \tag{28}$$

$$h_v^{(\ell+1)} = \sum_{u \in G} m_{vu}. \tag{29}$$

Denote the feature dimension of layer $\ell$ by $d_\ell$. Hence we have $h_u^{(\ell)} \in \mathbb{R}^\ell$ and $\phi(h_v^{(\ell)}, h_u^{(\ell)}) \in \mathbb{R}^{d_{\ell+1}}$.

As shorthand notation, we collect the latent representations in layer $(\ell)$ into a feature matrix $X^\ell$ whose columns are given as

$$[X^\ell]_{u:} = h_u^{(\ell)}. \tag{30}$$

We then define the map $\Phi : \mathbb{R}^{N \times d_\ell} \to \mathbb{R}^{N \times N \times d_{\ell+1}}$ componentwise as

$$[\Phi(X^\ell)]_{vu:} = \phi([X^\ell]_{v:}, [X^\ell]_{u:}) \equiv \phi(h_v^\ell, h_u^\ell). \tag{31}$$

With this we find

$$[X^{\ell+1}]_{v:} = \sum_u [\psi(L)]_{vu} \ [\Phi(X^\ell)]_{vu:}. \tag{32}$$

This now allows us to prove:

**Theorem B.2.** *Suppose we are given two graphs $G_1, G_2$ defined on a common node set. Denote the collection of node-wise latent representations generated by a $K$-layer deep Laplace transform based message passing network $\Xi_{\psi,\phi}$ by $X_1^K, X_2^K$ respectively.*

*Suppose there is a constant $C$ so that $C \geq \|\psi(L_1)\|, \|\psi(L_2)\|$. Assume that the message function $\phi$ is Lipschitz contiuous:*

$$\|\phi([X_1]_{v:}, [X_1]_{u:}) - \phi([X_2]_{v:}, [X_1]_{u:})\| \leq \frac{L}{4} \left( \|[X_1]_{v:}, -[X_2]_{v:}\| + \|[X_1]_{u:}, -[X_2]_{u:}\| \right). \tag{33}$$

*Further asume that $\Phi$ is bounded, with $\|\Phi(\cdot)\| \leq D$. Then there is a constant $C_\Xi$ so that*

$$\|X_1^K - X_2^K\| \leq C_\Xi \cdot \|\psi(L_1) - \psi(L_2)\|. \tag{34}$$

*Proof.* We note:

$$[X_1^K]_{v:} - [X_2^K]_{v:} = \sum_u [\psi(L_1)]_{vu} \ [\Phi(X_1^{K-1})]_{vu:} - \sum_u [\psi(L_2)]_{vu} \ [\Phi(X_2^{K-1})]_{vu:} \tag{35}$$

$$= \left[ \sum_u \left( [\psi(L_1)]_{vu} - \psi(L_2)]_{vu} \right) \ [\Phi(X_1^{K-1})]_{vu:} \right] + \left[ \sum_u [\psi(L_2)]_{vu} \left( [\Phi(X_1^{K-1})]_{vu:} - [\Phi(X_2^{K-1})]_{vu:} \right) \right]. \tag{36}$$

With this, we find

$$\|X_1^K - X_2^K\| \tag{37}$$

$$\leq \left( \sum_v \left\| \sum_u \left( [\psi(L_1)]_{vu} - \psi(L_2)]_{vu} \right) \ [\Phi(X_1^{K-1})]_{vu:} \right\|^2 \mu_v \right)^{\frac{1}{2}} \tag{38}$$

$$+ \left( \sum_v \left\| \sum_u [\psi(L_2)]_{vu} \left( [\Phi(X_1^{K-1})]_{vu:} - [\Phi(X_2^{K-1})]_{vu:} \right) \right\|^2 \mu_v \right)^{\frac{1}{2}} \tag{39}$$

with the node weights (c.f. Section A.1) denoted by $\{\mu_v\}_v$.

To bound the first term, we note:

$$\left( \sum_v \left\| \sum_u \left( [\psi(L_1)]_{vu} - \psi(L_2)]_{vu} \right) \ [\Phi(X_1^{K-1})]_{vu:} \right\|^2 \mu_v \right)^{\frac{1}{2}} \tag{40}$$

$$\leq D \cdot \left( \sum_{u,v} |[\psi(L_1)]_{vu} - \psi(L_2)]_{vu}|^2 \mu_v \right)^{\frac{1}{2}} \tag{41}$$

$$\lesssim D \|\psi(L_1) - \psi(L_2)\| \cdot \left( \sum_v \mu_v \right)^{\frac{1}{2}} \tag{42}$$

$$= D \sqrt{\mu(G)} \|\psi(L_1) - \psi(L_2)\|, \tag{43}$$

Here we have bounded $|[\psi(L_1)]_{vu} - \psi(L_2)]_{vu}| \lesssim \|\psi(L_1) - \psi(L_2)\|$.[1]

Let us next bound the second term. We have

$$\left( \sum_v \left\| \sum_u [\psi(L_2)]_{vu} \left( [\Phi(X_1^{K-1})]_{vu:} - [\Phi(X_2^{K-1})]_{vu:} \right) \right\|^2 \mu_v \right)^{\frac{1}{2}} \tag{44}$$

$$\leq C \cdot \left( \sum_{u,v} \left\| \left( [\Phi(X_1^{K-1})]_{vu:} - [\Phi(X_2^{K-1})]_{vu:} \right) \right\|^2 \mu_v \mu_u \right)^{\frac{1}{2}} \tag{45}$$

$$\leq CL \left( \sum_u \mu_u \right)^{\frac{1}{2}} \cdot \left( \sum_u \|[X_1^{k-1}]_{u:}, -[X_2^{K-1}]_{u:}\|^2 \mu_u \right)^{\frac{1}{2}}. \tag{46}$$

$$= CL\sqrt{\mu(G)} \cdot \|X_1^{K-1} - X_2^{K-1}\|. \tag{47}$$

Putting the two terms together yields the iteration step

$$\|X_1^K - X_2^K\| \leq C_A \|\psi(L_1) - \psi(L_2)\| + C_B \|X_1^{K-1} - X_2^{K-1}\|, \tag{48}$$

with constants $C_A, C_B$ as determined above. Iterating this through to $K = 0$ yields the following:

Let $d_K := \|X_1^K - X_2^K\|$ and $\Delta := \|\psi(L_1) - \psi(L_2)\|$. Assume the recursive bound

$$d_K \leq C_A \Delta + C_B \, d_{K-1}, \qquad K \geq 1.$$

Furthermore we have the intial condition $d_0 = \|X_1^0 - X_2^0\| = \|X - X\| = 0$.

Iterating the inequality yields

$$\begin{aligned} d_K &\leq C_A \Delta + C_B d_{K-1} \\ &\leq C_A \Delta + C_B \left( C_A \Delta + C_B d_{K-2} \right) \\ &= C_A \Delta (1 + C_B) + C_B^2 d_{K-2} \\ &\vdots \\ &\leq C_A \Delta \sum_{j=0}^{K-1} C_B^j + C_B^K d_0. \end{aligned}$$

Using $d_0 = 0$, we obtain

$$\|X_1^K - X_2^K\| \leq C_A \|\psi(L_1) - \psi(L_2)\| \sum_{j=0}^{K-1} C_B^j.$$

Evaluating the geometric series gives the closed-form bound

$$\|X_1^K - X_2^K\| \leq \begin{cases} C_A \|\psi(L_1) - \psi(L_2)\| \dfrac{1 - C_B^K}{1 - C_B}, & C_B \neq 1, \\[2mm] K \, C_A \|\psi(L_1) - \psi(L_2)\|, & C_B = 1. \end{cases}$$

In particular, if $0 \leq C_B < 1$, then

$$\|X_1^K - X_2^K\| \leq \frac{C_A}{1 - C_B} \|\psi(L_1) - \psi(L_2)\|,$$

which provides a uniform bound independent of $K$.

$\square$

---

[1] We are using the standard notation '$\lesssim$' to denote smaller than, up to a fixed 'universal' constant.

Combining the above Theorem with (27) then yields

**Corollary B.3.** *In the setting of Theorem B.2 we have*

$$\|X_1^K - X_2^K\| \leq C_\Xi \cdot \int_0^\infty \hat{\psi}(t) \left\| e^{-tL_1} - e^{-tL_2} \right\| dt = C_\Xi \max_i \{ \int_0^\infty \hat{\psi}_i(t) \left\| e^{-tL_1} - e^{-tL_2} \right\| dt \}.$$

$$(49)$$

Here the last equality follows since we are taking the maximum over all used propagation matries, and only ever use a single one.

Hence we have established a bound on variations of node-wise latent embeddings in terms of the heat kernel distance $\left\| e^{-tL_1} - e^{-tL_2} \right\|$ for spectral Laplace transform based methods.

## B.2 GRAPH LEVEL STABILITY

Next we promote the node-level stability results we derived in Subsection B.1 above to graph level stability results:

We may summarize Theorems B.1 & B.2 as: Given an input node-feature matrix $X$ on graphs $G_1, G_2$ sharing a common node set, there exists cosntants $C$ so that the output variation $\|X_1^K - X_2^K\|$ of $K$-layer deep Laplace transform based GNNs may be bounded as

$$\|X_1^K - X_2^K\| \leq C \max_i \left\{ \int_0^\infty \hat{\psi}_i(t) \| e^{-tL_1} - e^{tL_2} \| dt \right\}.$$

$$(50)$$

We are now interested in how these stability results translate to graph level features. To this end we first specify our graph level aggregation method $\Omega$:

**Definition B.2.** At the final ($K^{\text{th}}$-)layer We aggregate node-level embeddings $X^K \in \mathbb{R}^{N \times d_K}$ of individual nodes to graph-embeddings $\Omega(X) \in \mathbb{R}^{d_K}$ as $\Omega(X^K)_j = \sum_{i=1}^N |X_{ij}^K| \cdot \mu_i$. Here $\{\mu_i\}_i$ is the set of node-weights.

Graph level features $F_1, F_2$ are then generates as $F_1 = \Omega(X_1^K)$ and $F_2 = \Omega(X_2^K)$.

With this we find:

**Theorem B.4.** *In the setting of (50), with graph level aggregation as in Definition B.2, we find*

$$\|F_1 - F_2\| \leq C \max_i \left\{ \int_0^\infty \hat{\psi}_i(t) \| e^{-tL_1} - e^{tL_2} \| dt \right\}.$$

$$(51)$$

*Proof.* We only have to note that the aggregation method $\Omega$ is 1-Lipschitz (as a consequence of the reverse triangle inequality). Hence

$$\|F_1 - F_2\| = \|\Omega(X_1^K) - \Omega(X_2^K)\| \leq \|X_1^K - X_2^K\|$$

$$(52)$$

and the claim follows from 50. $\qquad\square$

## C STABILITY OF LATENT REPRESENTATIONS GENERATED BY LAPLACE-TRANSFORM BASED GNNS FOR GRAPHS ON DIFFERING RESOLUTION SCALES

In this section, we establish bounds on differences in latent representations generated for Laplace-transform based GNNs when confronted with graphs describing the same underlying object at multiple resolutions. We briefly recall the setting:

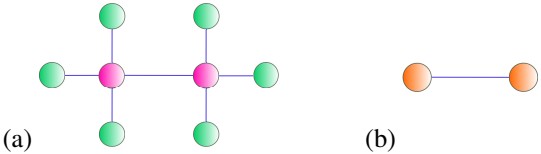

Figure 4: (a) Original graph $G$ (b) Coarse grained $\underline{G}$

We have a high resolution graph $G$ with associated Laplacian $L$ and node-feature matrix $X$. We also have a lower resolution graph $\underline{G}$, with associated Laplacian $\underline{L}$ and node-feature matrix $\underline{X} := J^\downarrow X$ arising from the original node feature matrix $X$ on $G$ via projection to $\underline{G}$. Here we have made use of the projection opreator $J^\downarrow$ which averages node information over clusters that are condensed into supernodes. Interpolation $J^\uparrow$ instead assings information at each supernode in $\underline{G}$ to every node making up the corresponding cluster in $G$.

We have (Koke, 2026) (c.f. also Appendix E) that $J^\downarrow J^\uparrow = I_{\underline{G}}$ (i.e. equal to the identity matrix on $\underline{G}$).

## C.1 NODE LEVEL RESULTS

We are then initially interested in the following question: Suppose we have a node feature matrix $X$ on the graph $G$. We can generate node-level latent embeddings $\Xi(X)$ by feeding this node-wise information into a (Laplace-transform based) node-level GNN $\Xi$. How different is this outcome from projecting $X$ onto the coarser graph $\underline{G}$, running the GNN there to generate node-level embeddings $\Xi(J^\downarrow X)$ and then interpolating the generated embeddings back up to $G$ via the interpolation operator $J^\uparrow$. That is to say, we are interested in the difference $\|\Xi(X) - J^\uparrow \Xi(J^\downarrow X)\|$.

### C.1.1 SPECTRAL LAPLACE-TRANSFORM METHODS

We begin by establishing $\|\Xi(X) - J^\uparrow \Xi(J^\downarrow X)\| \leq C \cdot \max_i \left\{ \int_0^\infty |\hat{\psi}_i(t)| \cdot \|e^{-tL_\omega} - J^\uparrow e^{-t\underline{L}} J^\downarrow\| \right\}$ for spectral methods:

**Theorem C.1.** Let $\Xi_{\mathscr{W}, \mathscr{B}, \Psi}$ be a $K$-layer deep Laplace transform based network. Assume that $\sum_{i \in I} \|W_i^\ell\| \leq W$ and $\|B^k\| \leq B$. Choose $C \geq \|\Psi_i(L)\|, \|\Psi_i(\tilde{L})\|$ ($i \in I$) and w.l.o.g. assume $CW > 1$. Set $\max_{i \in I}\{\|\psi_i(L) - J^\uparrow \psi_i(\underline{L}) J^\downarrow\|\} = \delta$. With this, we have that

$$\|\Xi_{\mathscr{W}, \mathscr{B}, \Psi}(X) - J^\uparrow \Xi_{\mathscr{W}, \mathscr{B}, \Psi}(J^\downarrow X)\| \leq \left[ K \cdot C^K W^{K-1} \cdot \left( \|X\| + \frac{1}{CW - 1} B \right) \right] \cdot \delta. \tag{53}$$

*Proof.* Let us define

$$\underline{X} := J^\downarrow X. \tag{54}$$

Let us further use the notation $\underline{\psi}_i := \psi_i(\underline{L})$ and $\psi_i := \psi_i(L)$.

Denote by $X^\ell$ and $\underline{X}^\ell$ the (hidden) feature matrices generated in layer $\ell$ for networks based on $\psi_i$ and $\underline{\psi}_i$ respectively: I.e. we have

$$X^\ell = \rho \left( \sum_{i \in I} \psi_i X^{\ell-1} W_i^\ell + B^\ell \right) \tag{55}$$

and

$$\underline{X}^\ell = \rho \left( \sum_{i \in I} \underline{\psi}_i \underline{X}^{\ell-1} W_i^\ell + \underline{B}^\ell \right). \tag{56}$$

Since bias terms are proportional to constant vectors on the graphs, we have

$$J^\downarrow B = \underline{B} \tag{57}$$

and

$$J^\uparrow \underline{B} = B \tag{58}$$

for bias matrices $B$ and $\underline{B}$ in networks deployed on $G$ and $\underline{G}$ respectively.

We then have

$$\|\Xi_{\mathscr{W},\mathscr{B},\Psi}(X) - J^{\uparrow}\Xi_{\mathscr{W},\mathscr{B},\Psi}(J^{\downarrow}X)\| \tag{59}$$

$$=\|X^K - J^{\uparrow}\underline{X}^K\| \tag{60}$$

$$= \left\|\rho\left(\sum_{i\in I}\psi_i X^{K-1}W_i^K + B^K\right) - J^{\uparrow}\rho\left(\sum_{i\in I}\underline{\psi}_i\underline{X}^{K-1}W_i^K + \underline{B}^L\right)\right\| \tag{61}$$

$$= \left\|\rho\left(\sum_{i\in I}\psi_i X^{K-1}W_i^K + B^K\right) - \rho\left(J^{\uparrow}\sum_{i\in I}\tilde{\psi}_i\underline{X}^{K-1}W_i^K + B^L\right)\right\| \tag{62}$$

$$\tag{63}$$

Here we used the fact that $\rho$ and $J^{\uparrow}$ commute (since $J^{\uparrow}$ is an assignment matrix and has all non-zero entries equal to one). Using the fact that the non-linearity $\rho(\cdot)$ is 1-Lipschitz-continuous we can establish

$$\|\Xi_{\mathscr{W},\mathscr{B},\Psi}(X) - J^{\uparrow}\Xi_{\mathscr{W},\mathscr{B},\Psi}(J^{\downarrow}X)\| \tag{64}$$

$$\leq \left\|\rho\left(\sum_{i\in I}\psi_i X^{K-1}W_i^K + B^K\right) - \rho\left(J^{\uparrow}\sum_{i\in I}\underline{\psi}_i\underline{X}^{K-1}W_i^K + B^K\right)\right\| \tag{65}$$

$$\leq \left\|\sum_{i\in I}\psi_i X^{K-1}W_i^K + B^K - J^{\uparrow}\sum_{i\in I}\underline{\psi}_i\underline{X}^{K-1}W_i^K + B^K\right\|. \tag{66}$$

Since $J^{\downarrow}J^{\uparrow} = Id_{\underline{G}}]$, we have

$$\|\Xi_{\mathscr{W},\mathscr{B},\Psi}(X) - J^{\uparrow}\Xi_{\mathscr{W},\mathscr{B},\Psi}(J^{\downarrow}X)\| \tag{67}$$

$$\leq \left\|\sum_{i\in I}\psi_i X^{K-1}W_i^K - \sum_{i\in I}(J^{\uparrow}\underline{\psi}_i J^{\downarrow})J^{\uparrow}\underline{X}^{K-1}W_i^K\right\| \tag{68}$$

From this, we find (using $\|J^{\uparrow}\|, \|J^{\downarrow}\| \leq 1$ ), that

$$\|\Xi_{\mathscr{W},\mathscr{B},\Psi}(X) - J^{\uparrow}\Xi_{\mathscr{W},\mathscr{B},\Psi}(J^{\downarrow}X)\| \tag{69}$$

$$\leq \left\|\sum_{i\in I}\psi_i X^{K-1}W_i^K - \sum_{i\in I}(J^{\uparrow}\underline{\psi}_i J^{\downarrow})J^{\uparrow}\underline{X}^{K-1}W_i^K\right\| \tag{70}$$

$$\leq \left\|\sum_{i\in I}(\psi_i - J^{\uparrow}\underline{\psi}_i J^{\downarrow})X^{K-1}W_i^K\right\| + \sum_{i\in I}\|J^{\uparrow}\underline{\psi}_i J^{\downarrow}\| \cdot \|J^{\uparrow}\underline{X}^{K-1} - X^{K-1}\| \cdot \|W_i^K\| \tag{71}$$

$$\leq \left\|\sum_{i\in I}(\psi_i - J^{\uparrow}\underline{\psi}_i J^{\downarrow})X^{K-1}W_i^K\right\| + CW \cdot \|J^{\uparrow}\underline{X}^{K-1} - X^{K-1} \tag{72}$$

$$\leq \sum_{i\in I}\left\|(\psi_i - J^{\uparrow}\underline{\psi}_i J^{\downarrow})\right\| \cdot \left\|X^{K-1}\right\| \cdot \left\|W_i^K\right\| + CW \cdot \|\tilde{J}^{\downarrow}\underline{X}^{K-1} - X^{K-1}\| \tag{73}$$

$$\leq \delta \cdot \left\|X^{K-1}\right\|W + CW \cdot \|\tilde{J}^{\uparrow}\underline{X}^{K-1} - X^{K-1}\| \tag{74}$$

$$\tag{75}$$

Arguing as in the proof of Theorem B.1 then yields the claim.

$$\square$$

### C.1.2   MESSAGE PASSING BASED LAPLACE TRANSFORM METHODS:

We next establish similar results for message passing networks. Sticking with the notation of the preceding Subsection C.1.1, we are hence interested in bounding $X^K - J^{\uparrow}\underline{X}^K$, with $\underline{X}$ being the node feature matrix generated after a $K$-layer Laplace transform based MPNN.

**Theorem C.1.** *Assume there is a constant $C$ so that $C \geq \|\psi(L)\|, \|\psi(\underline{L})\|$. Assume that the message function $\phi$ is Lipschitz contiuous:*

$$\|\phi([X_1]_{v:}, [X_1]_{u:}) - \phi([X_2]_{v:}, [X_1]_{u:})\| \leq \frac{L}{4} \left( \|[X_1]_{v:}, -[X_2]_{v:}\| + \|[X_1]_{u:}, -[X_2]_{u:}\| \right). \quad (76)$$

*Further asume that $\Phi$ is bounded, with $\|\Phi(\cdot)\| \leq D$. Then there is a constant $C_\Xi$ so that*

$$\|X^K - J^\uparrow \underline{X}^K\| \leq C_\Xi \cdot \int_0^\infty \hat{\psi}(t) \|e^{-tL} - J^\uparrow e^{-t\underline{L}} J^\downarrow\|. \quad (77)$$

*Proof.* We note:

$$X_{v:}^K - [J^\uparrow \underline{X}^K]_{v:} = X_{v:}^K - \sum_{a \in \underline{G}} J_{va}^\uparrow \underline{X}_{a:}^K = \sum_{u \in G} [\psi(L)]_{vu} [\Phi(X^{K-1})]_{vu:} - \sum_{a,b \in \underline{G}} J_{va}^\uparrow [\psi(\underline{L})]_{ab} [\Phi(\underline{X}^{K-1})]_{ab:}$$
$$(78)$$

Creatively adding zero yields

$$X_{v:}^K - [J^\uparrow \underline{X}^K]_{v:} \qquad (79)$$
$$= \sum_{u \in G} [\psi(L)]_{vu} [\Phi(X^{K-1})]_{vu:} - \sum_{u \in G} [\psi(L)]_{vu} [\Phi(J^\uparrow \underline{X}^{K-1})]_{vu:} \qquad (80)$$
$$+ \sum_{u \in G} [\psi(L)]_{vu} [\Phi(J^\uparrow \underline{X}^{K-1})]_{vu:} - \sum_{a,b \in \underline{G}} J_{va}^\uparrow [\psi(\underline{L})]_{ab} [\Phi(\underline{X}^{K-1})]_{ab:}. \qquad (81)$$

For the first term, we note

$$\left( \sum_{v \in G} \left\| \sum_{u \in G} [\psi(L)]_{vu} [\Phi(X^{K-1})]_{vu:} - \sum_{u \in G} [\psi(L)]_{vu} [\Phi(J^\uparrow \underline{X}^{K-1})]_{vu:} \right\|^2 \mu_v \right)^{\frac{1}{2}} \qquad (82)$$
$$\leq C \cdot \left( \sum_{v,u \in G} \sum_{u \in G} \left\| [\Phi(X^{K-1})]_{vu:} - [\Phi(J^\uparrow \underline{X}^{K-1})]_{vu:} \right\|^2 \mu_v \right)^{\frac{1}{2}} \qquad (83)$$
$$\leq CL \left( \sum_u \mu_u \right)^{\frac{1}{2}} \cdot \left( \sum_u \| [X^{K-1}]_{u:}, -[J^\uparrow \underline{X}]_{u:} \|^2 \mu_u \right)^{\frac{1}{2}}. \qquad (84)$$
$$= CL \sqrt{\mu(G)} \cdot \| X^{K-1} - J^\uparrow \underline{X}^{K-1} \|. \qquad (85)$$

Hence let us consider the second term. By Lemma C.2 below, we then have

$$\sum_{v \in G} \mu_g \left\| \sum_{u \in G} [\psi(L)]_{vu} [\Phi(J^\uparrow \underline{X}^{K-1})]_{vu:} - \sum_{a,b \in \underline{G}} J_{va}^\uparrow [\psi(\underline{L})]_{ab} [\Phi(\underline{X}^{K-1})]_{ab:} \right\|^2 \tag{86}$$

$$= \sum_{v \in G} \mu_g \left\| \sum_{u \in G} [\psi(L)]_{vu} \sum_{a,b \in \underline{G}} J_{va}^\uparrow J_{ub}^\uparrow [\Phi(\underline{X}^{K-1})]_{ab:} - \sum_{a,b \in \underline{G}} J_{va}^\uparrow [\psi(\underline{L})]_{ab} [\Phi(\underline{X}^{K-1})]_{ab:} \right\|^2 \tag{87}$$

$$= \sum_{v \in G} \mu_g \left\| \sum_{a,b \in \underline{G}} J_{va}^\uparrow [\psi(L) J^\uparrow]_{vb} [\Phi(\underline{X}^{K-1})]_{ab:} - \sum_{a,b \in \underline{G}} J_{va}^\uparrow [\psi(\underline{L})]_{ab} [\Phi(\underline{X}^{K-1})]_{ab:} \right\|^2 \tag{88}$$

$$= \sum_{v \in G} \mu_g \left\| \sum_{a,b \in \underline{G}} J_{va}^\uparrow [\psi(L) J^\uparrow]_{vb} [\Phi(\underline{X}^{K-1})]_{ab:} - \sum_{a,b \in \underline{G}} J_{va}^\uparrow [J^\uparrow \psi(\underline{L})]_{vb} [\Phi(\underline{X}^{K-1})]_{ab:} \right\|^2 \tag{89}$$

$$\leq D^2 \sum_{v \in G} \mu_g \left( \sum_{a,b \in \underline{G}} J_{va}^\uparrow | [\psi(L) J^\uparrow]_{vb} - [J^\uparrow \psi(\underline{L})]_{vb} | \right)^2 \tag{90}$$

$$\lesssim D^2 \sum_{v \in G} \mu_g \left( \sum_{a \in \underline{G}} J_{va}^\uparrow \| \psi(L) J^\uparrow - J^\uparrow \psi(\underline{L}) \| \right)^2 \tag{91}$$

$$\leq D^2 \mu(G) \| \psi(L) J^\uparrow - J^\uparrow \psi(\underline{L}) \|^2 \tag{92}$$

$$\lesssim D^2 \mu(G) \| \psi(L) - J^\uparrow \psi(\underline{L}) J^\downarrow \|^2. \tag{93}$$

Here the third step follows from the fact that $J_{va}^\uparrow [J^\uparrow \psi(\underline{L})]_{vb} = J_{va}^\uparrow [\psi(\underline{L})]_{ab}$. The last step follows from the fact that $J^\downarrow$ is surjective.

Recursively, we thus have established that

$$\| X^K - J^\uparrow \underline{X}^K \| \leq C_A \| \psi(L) - J^\uparrow \psi(\underline{L}) J^\downarrow \| + C_B \| X^{K-1} - J^\uparrow \underline{X}^{K-1} \|. \tag{94}$$

We may unroll the recursion to obtain

$$\| X^K - J^\uparrow \underline{X}^K \| \leq C_A \sum_{i=0}^{K-1} C_B^i \| \psi(L) - J^\uparrow \psi(\underline{L}) J^\downarrow \| + C_B^K \| X^0 - J^\uparrow \underline{X}^0 \|. \tag{95}$$

If $C_B \neq 1$, the geometric sum yields

$$\| X^K - J^\uparrow \underline{X}^K \| \leq C_A \frac{1 - C_B^K}{1 - C_B} \| \psi(L) - J^\uparrow \psi(\underline{L}) J^\downarrow \| + C_B^K \| X^0 - J^\uparrow \underline{X}^0 \|. \tag{96}$$

Finally we recall from (3) that the initial input $X^0$ into the Laplace-transform message passing layers arises from the input features $X$ via an initial Laplace-transform propagation step as

$$X^0 = \psi(L) X. \tag{97}$$

Thus we also find

$$J^\uparrow \underline{X}^0 = J^\uparrow \psi(\underline{L}) [J^\downarrow X]. \tag{98}$$

From this we may infer that

$$\| X^0 - J^\uparrow \underline{X}^0 \| = \| \psi(L) X - J^\uparrow \psi(\underline{L}) J^\downarrow X \| \leq \| \psi(L) - J^\uparrow \psi(\underline{L}) \| \cdot \| X \|. \tag{99}$$

Hence both summands in (96) may be bounded in terms of $\| \psi(L) - J^\uparrow \psi(\underline{L}) \|$. Using

$$\| \psi(L) - J^\uparrow \psi(\underline{L}) \| \leq \int_0^\infty \hat{\psi}(t) \| e^{-tL} - J^\uparrow e^{-t\underline{L}} J^\downarrow \| dt, \tag{100}$$

we have thus established the claim. $\qquad \square$

Finally we provide the Lemma used in the proof above:

**Lemma C.2.** *We have*

$$[\Phi(J^\uparrow \underline{X}^\ell)]_{vu:} = \sum_{a,b \in \underline{G}} J^\uparrow_{va} J^\uparrow_{ub} [\Phi(\underline{X}^\ell)]_{ab:}. \tag{101}$$

*Proof.* Indeed, this follows because entries of $J^\uparrow$ satisfy $J^\uparrow_{ua} \in \{0,1\}$ and every node in $G$ gets assigned information exactly from only one node in $\underline{G}$. Thus the sum $\sum_{a,b \in \underline{G}} J^\uparrow_{va} J^\uparrow_{ub} [[\Phi(\underline{X}^\ell)]_{ab:}]$ contains precisely *one* non-zero summand. This summand in turn, corresponds to those indices $a, b \in \underline{G}$ which are mapped to $v$ and $u$ respectively under $J^\uparrow$. Since entries in $J^\uparrow$ are binary (and hence do not diminish or blow up signal if they are non-zero) the claim follows. $\square$

## C.2 GRAPH LEVEL RESULTS AND PROOF OF THEOREM 3.1

It remains to translate the results of Theorems C.1 & C.1 to the graph level. To this end, we first note the following:

**Lemma C.2.** *With the graph level aggregation function $\Omega$ as in Definition B.2, we have*

$$\Omega(\underline{X}) = \Omega(J^\uparrow \underline{X}). \tag{102}$$

*Proof.* This fact follows immediately from the fact that $J^\uparrow$ assigns the information present at the supernode $a \in \underline{G}$ to all nodes $u$ in the cluster within $G$ that correspond to the supernode a, together with the fact that the node-weight $\mu_a$ of the node $a \in \underline{G}$ is the sum over all of the node weights $\mu_u$ of nodes $u$ in the cluster within $G$ that corresponds to the supernode $a$ under coarse graining (c.f. also (117) in Appendix E). $\square$

Using the above lemma, we find

$$\|\underline{F} - F\| = \|\Omega(\underline{X}^K) - \Omega(X^K)\| = \|\Omega(J^\uparrow \underline{X}^K) - \Omega(X^K)\|. \tag{103}$$

Using the 1-Lipschitzness of the aggregation function $\Omega$ then yields

$$\|\Omega(J^\uparrow \underline{X}^K) - \Omega(X^K)\| \leq \|J^\uparrow \underline{X}^K - X^K\|. \tag{104}$$

Together with Theorems C.1 & C.1 we have thus proved:

**Theorem C.3.** *Let $F$ and $\underline{F}$ be the latent embeddings generated for a graph $G$ its coarsified version $\underline{G}$ by a (spectral or message passing) network employing Laplace transform propagation. With $\{\Psi_i(L) = \int_0^\infty \hat{\psi}_i(t)e^{-tL}dt\}_{i \in I}$ the collection of all Laplace-transform propagation matrices used within the network, we have $\|F - \underline{F}\| \leq C \cdot \max_i \left\{ \int_0^\infty |\hat{\psi}_i(t)| \cdot \|e^{-tL_\omega} - J^\uparrow e^{-t\underline{L}} J^\downarrow\| \right\}$.*

Together with (112) we have hene proved Theorem 3.1.

# D ADDITIONAL EXPERIMENTAL CONSIDERATIONS

## D.1 ADDITIONAL DETAILS ON INITIAL COARSE GRAINING EXPERIMENT

**Dataset:** The dataset we consider is the **QM7** dataset, introduced in Blum & Reymond (2009); Rupp et al. (2012). This dataset contains descriptions of 7165 organic molecules, each with up to seven heavy atoms, with all non-hydrogen atoms being considered heavy. A molecule is represented by its Coulomb matrix $C^{\text{Clmb}}$, whose off-diagonal elements

$$C_{ij}^{\text{Clmb}} = \frac{Z_i Z_j}{|R_i - R_j|} \tag{105}$$

correspond to the Coulomb-repulsion between atoms $i$ and $j$. We discard diagonal entries of Coulomb matrices; which would encode a polynomial fit of atomic energies to nuclear charge Rupp et al. (2012).

For each atom in any given molecular graph, the individual Cartesian coordinates $R_i$ and the atomic charge $Z_i$ are (in principle) also accessible individually. To each molecule an atomization energy - calculated via density functional theory - is associated. The objective is to predict this quantity. The performance metric is mean absolute error. Numerically, atomization energies are negative numbers in the range $-600$ to $-2200$. The associated unit is [*kcal/mol*].

**Details on collapsing procedure:**    Again, we make use of the QM7 dataset Rupp et al. (2012) and its Coulomb matrix description

$$C_{ij}^{\text{Clmb}} = \frac{Z_i Z_j}{|R_i - R_j|} \tag{106}$$

of molecules. We modify (all) molecular graphs in QM7 by deflecting hydrogen atoms (H) out of their equilibrium positions towards the respective nearest heavy atom. This is possible since the QM7 dataset also contains the Cartesian coordinates of individual atoms. Edge weights between heavy atoms then remain the same, while Coulomb repulsions between H-atoms and respective nearest heavy atom increasingly diverge; as is evident from (106).

Given an original molecular graph $G$ with node weights $\mu_i = Z_i$, the corresponding limit graph $\underline{G}$ corresponds to a coarse grained description, where heavy atoms and surrounding H-atoms are aggregated into single super-nodes.

Mathematically, $\underline{G}$ is obtained by removing all nodes corresponding to H-atoms from $G$, while adding the corresponding charges $Z_H = 1$ to the node-weights of the respective nearest heavy atom. Charges in (106) are modified similarly to generate the weight matrix $\underline{W}$.

On original molecular graphs, atomic charges are provided via one-hot encodings. For the graph of methane – consisting of one carbon atom with charge $Z_C = 6$ and four hydrogen atoms of charges $Z_H = 1$ – the corresponding node-feature-matrix is e.g. given as

$$X = \begin{pmatrix} 0 & 0 & \cdots & 0 & 1 & 0\cdots \\ 1 & 0 & \cdots & 0 & 0 & 0\cdots \\ 1 & 0 & \cdots & 0 & 0 & 0\cdots \\ 1 & 0 & \cdots & 0 & 0 & 0\cdots \\ 1 & 0 & \cdots & 0 & 0 & 0\cdots \end{pmatrix} \tag{107}$$

with the non-zero entry in the first row being in the 6$^{\text{th}}$ column, in order to encode the charge $Z_C = 6$ for carbon.

The feature vector of an aggregated node represents charges of the heavy atom and its neighbouring H-atoms jointly.

Node feature matrices are translated as $\underline{X} = J^{\downarrow} X$. Applying $J^{\downarrow}$ to one-hot encoded atomic charges yields (normalized) bag-of-word embeddings on $\underline{G}$: Individual entries of feature vectors encode how much of the total charge of the super-node is contributed by individual atom-types. In the example of methane, the limit graph $\underline{G}$ consists of a single node with node-weight

$$\mu = 6 + 1 + 1 + 1 + 1 = 10. \tag{108}$$

The feature matrix

$$\underline{X} = J^{\downarrow} X \tag{109}$$

is a single row-vector given as

$$\underline{X} = \left( \frac{4}{10}, 0, \cdots, 0, \frac{6}{10}, 0, \cdots \right). \tag{110}$$

**Experimental Setup:**    We randomly select 1500 molecules for testing and train on the remaining graphs. On QM7 we run experiments for 5 different random random seeds and report mean and standard deviation. All experiments were performed on a single NVIDIA Quadro RTX 8000 graphics card.

**Additional details on training and models:**    Typical GNN models are divided into **standard** architectures (GCN (Kipf & Welling, 2017), ChebNet (Defferrard et al., 2016), GATv2 (Brody et al., 2022)) GIN ((Xu et al., 2019) (together with its edge-weight enabled version (Hu et al., 2020) where

applicable)) and **multi- scale** architectures (UFGNet (Zheng et al., 2021), Lanczos (Liao et al., 2019)). Apart from UFGNet (already acting as a **pooling** layer) we also consider self-attention-pooling (Lee et al., 2019); both acting on the final layer (SAG) and as acting on the output of each indivifual layer, with resulting layer-wise features concatenated to produce the final embedding (SAG-M). All considered convolutional layers are incorporated into a two layer deep and fully connected graph convolutional architecture. In each hidden layer, we set the width (i.e. the hidden feature dimension) to $F_1 = F_2 = 64$. For ChebNet, we set the polynomial order to $K = 2$ as is customary. Lanczos uses 20 Lanczos iterations, as proposed in the original paper (Liao et al., 2019). UFGNet uses Haar wavelets. For all baselines, the standard mean-aggregation scheme is employed after the graph-convolutional layers to generate graph level features. Finally, predictions are generated via an MLP.

For the Spectral$_{\text{Res.}}$-architecture, we set $\lambda = 1$ and and build filters using the $k = 1$ and $k = 2$ atoms in $\Psi^{\text{Res}} = \{(z + \lambda)^{-k}\}_{k \in \mathbb{N}}$. For MPNN$_{\text{Res.}}$ we use the $k = 1$ atom as $\psi$.

For the Spectral$_{\text{Exp.}}$-, we set $t = 1$ and and build filters using the $k = 1$ and $= 2$ atoms in $\Psi^{\text{Exp}} = \{e^{-(kt_0)z}\}_{k \in \mathbb{N}}$. For MPNN$_{\text{Res.}}$ we use the $k = 1$ atom as $\psi$.

As aggregation, we employ the graph level feature aggregation scheme discuseed in Definition B.2, with node weights set to atomic charges of individual atoms. Predictions are then generated via a final MLP with the same specifications as the one used for baselines.

### D.2 FURTHER DISCUSSIONS ON STABILITY RESULTS OF TABLE 2

We can further understand the small difference of latent embeddings using Theorem C.3. Let use use the shorthand notation

$$\eta(t) = \|e^{-Lt} - J^{\uparrow} e^{-t\underline{L}} J^{\downarrow}\|. \tag{111}$$

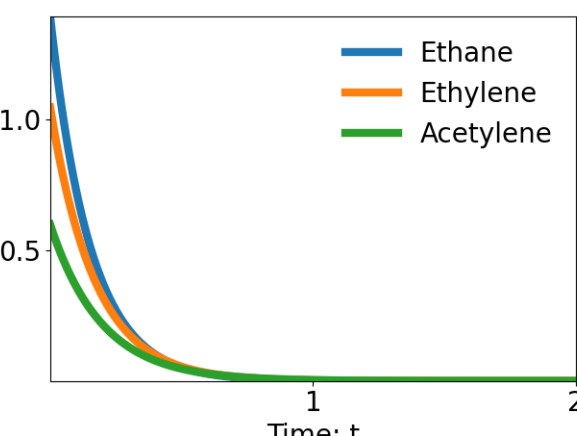

Figure 5: $\|e^{-Lt} - J^{\uparrow} e^{-t\underline{L}} J^{\downarrow}\|$ for molecules in QM7

Exemplarily considering exponential propagation matrices we have (with $t_k = k$) that $\int_0^{\infty} |\hat{\psi}_k(t)| \|e^{-tL} - J^{\uparrow} e^{-t\underline{L}} J^{\downarrow}\| dt = \|e^{-kL} - J^{\uparrow} e^{-k\underline{L}} J^{\downarrow}\|$, we thus have $\|F - \underline{F}\| \lesssim \max_{k \geq 1} |\eta(k)|$. Investigating the differences $\eta(t) = \|e^{-tL} - J^{\uparrow} e^{-t\underline{L}} J^{\downarrow}\|$ of diffusion flows, we find that $\eta(t)$ drops to zero fast, as exemplarily plotted in Fig. 5 for the first few molecules of QM7. In particular $|\eta(k)|_{k \geq 1} \approx 0$.

Using this as an upper bound in Theorem C.3 shows that embeddings $F, \underline{F}$ of graphs describing the same molecule at different resolution scales are similar. This explains the ability to generalize between scales.

### D.3   TRANSFERABILITY ON GRAPHS GENERATED VIA STOCHASTIC BLOCK MODELS

**Stochastic Block Models:**   Stochastic block models (Holland et al., 1983) are generative models for random graphs that produce graphs containing strongly connected communities. In our experiments in this section, we consider a stochastic block model whose distributions is characterized by four parameters: The number of communities $c_{\text{number}}$ determine how many (strongly connected) communities are present in the graph that is to be generated. The community size $c_{\text{size}}$ determines the number of nodes belonging to each (strongly connected) community. The probability $p_{\text{connect}}$ determines the probability that two nodes within the same community are connected by an edge. The probability $p_{\text{inter}}$ determines the probabilities that two nodes in *different* communities are connected by an edge.

**Experimental Setup:**   Since stochastic block models do not generate node-features, we equip each node with a randomly-generated unit-norm feature vector. Given such a graph $G$ drawn from a stochastic block model, we then compute a version $\underline{G}$ of this graph, where all communities are collapsed to single nodes. We then compare the feature vectors generated for $G$ and $\underline{G}$. All experiments were performed on a single NVIDIA Quadro RTX 8000 graphics card.

**Experiment: Varying the Connectivity within the Communities:**   We fix the parameters $c_{\text{number}}, c_{\text{size}}$ and $p_{\text{inter}}$ in our stochastic block model. We then vary the probability $p_{\text{connect}}$ that two nodes within the same community are connected by an edge from $p_{\text{connect}} = 0$ to $p_{\text{connect}} = 1$. This corresponds to varying the connectivity within the communities from very sparse (or in fact no connectivity) to full connectivity (i.e. the community being a clique). In Figure 6 below, we then plot the difference of feature vectors generated by models for $G$ and $\underline{G}$ respectively. For each $p_{\text{connect}} \in [0, 1]$, results are averaged over 100 graphs randomly drawn from the same stochastic block model.

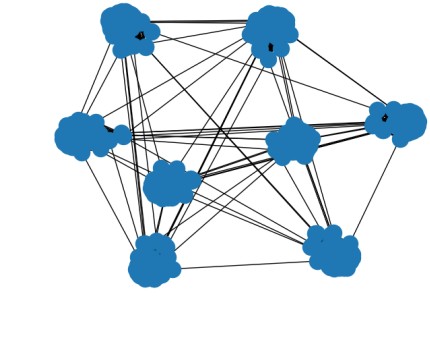
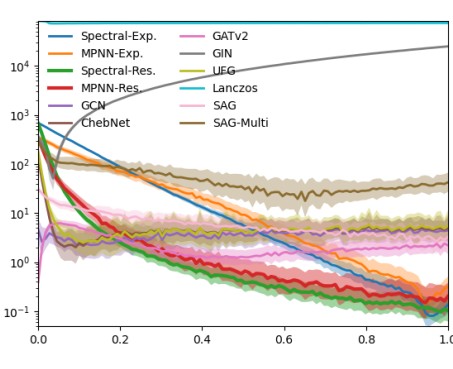

(a)                                (b)

Figure 6: (a) Example Graph (b) Varying the parameter $p_{\text{connect}} \in [0, 1]$ for fixed $c_{\text{size}} = 60$, $p_{\text{inter}} = 2/c_{\text{size}}^2$ and $c_{\text{number}} = 12$.

We have chosen $p_{\text{inter}} = 2/c_{\text{size}}^2$ so that – on average – *clusters* are connected by two edges. The choice of two edges (as opposed to $1, 3, 4, 5, ...$) between clusters is not important.

## E   FURTHER DISCUSSION OF THE SETTING OF COARSE-GRAINING GRAPHS

Let us consider graphs that contain clusters of nodes which are connected by significantly larger edge weights than those of edges outside of these clusters. From a diffusion perspective, information in a graph equalizes faster along edges with large weights.

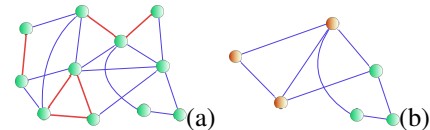

Figure 7: (a) $G$ (stongly connected) clusters in red (b) Coarse grained $\underline{G}$

In the limit where edge-weights within certain sub-graphs tend to infinity, information within these clusters equalizes immediately. Such clusters thus effectively behave as single nodes. We might thus consider a coarse grained graph $\underline{G}$ where strongly connected clusters are fused together and represented only via single nodes. This naturally leads to the notion of graph coarsification, as first formalized and studied in (Loukas & Vandergheynst, 2018; Loukas, 2019).

In our case at hand the node set $\underline{\mathcal{G}}$ of the coarse grained graph $\underline{G}$ is then given by the set of connected components in $G_{\text{cluster}}$ (c.f. Fig 8). Edges $\underline{\mathcal{E}}$ are given by elements $(R, P) \in \underline{\mathcal{G}} \times \underline{\mathcal{G}}$ with non-zero accumulated edge weight $\underline{W}_{RP} = \sum_{r \in R} \sum_{p \in P} W_{rp}$. Node weights in $\underline{G}$ are defined accordingly by aggregating as $\underline{\mu}_R = \sum_{r \in R} \mu_r$. To compare signals on these two graphs, we define intertwining operators $J^\downarrow, J^\uparrow$ transferring information between $G$ and $\underline{G}$: Let $x$ be a scalar graph signal and let $\mathbb{1}_R$ be the vector that has $1$ as entry for nodes $r \in R$ and is zero otherwise. Denote by $u_R$ the entry of $u$ at node $R \in \underline{\mathcal{G}}$. Projection $J^\downarrow$ is then defined component-wise by evaluation at node $R \in \underline{\mathcal{G}}$ as the average of $x$ over $R$: $(J^\downarrow x)_R = \langle \mathbb{1}_R, x \rangle / \underline{\mu}_R$. Going in the opposite direction, interpolation is defined as $J^\uparrow u = \sum_{R \in \underline{\mathcal{G}}} u_R \cdot \mathbb{1}_R$. In this setting, we have (c.f. the discussion below) that

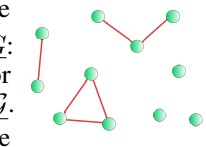

Figure 8: $G_{\text{cluster}}$

$$\|e^{-tL} - J^\uparrow e^{-t\underline{L}} J^\downarrow\| \lesssim 1/w_{\text{high}}^{\min} \to 0 \qquad \text{for any } t > 0. \tag{112}$$

Here $w_{\text{high}}^{\min} \gg 1$ denotes the minimal edge weight inside the strongly connected clusters in $G$. As the strength of the edge-weights in $G_{\text{cluster}}$ tends to infinity, we have by (112) that also $\eta(t) = \|e^{-Lt} - J^\uparrow e^{-\underline{L}t} J^\downarrow\| \to 0$ for any $t > 0$. Thus (for $t > 0$) the diffusion process $e^{-Lt}$ on $G$ acts essentially as first projecting the input-signal to $\underline{G}$ via $J^\downarrow$, then diffusing information on the coarse grained graph $\underline{G}$ via $e^{-\underline{L}t}$ and finally interpolating back to the original graph $G$ via $J^\uparrow$.

Following the proof in Koke (2026), we here illustrate

$$\|(L + Id)^{-1} - J^\uparrow (\underline{L} + Id)^{-1} J^\downarrow\| \lesssim 1/\lambda_1(L_{\text{high}}). \tag{113}$$

in this Appendix. Using (Koke, 2026, Corollary 3.18), this then also implies

$$\|e^{-tL} - J^\uparrow e^{-t\underline{L}} J^\downarrow\| \lesssim 1/w_{\text{high}}^{\min} \text{ for any } t > 0, \tag{114}$$

after noting the linear relation in scaling behaviour $\lambda_1(L_{\text{cluster}}) \sim w_{\text{high}}^{\min}$.

For convenience, we here explicitly restate the definitions leading up to this setting in full clarity and generality:

**Definition E.1.** Denote by $\underline{\mathcal{G}}$ the set of connected components in $G_{\text{high}}$. We give this set a graph structure as follows: Let $R$ and $P$ be elements of $\underline{\mathcal{G}}$ (i.e. connected components in $G_{\text{high}}$). We define the real number

$$\underline{W}_{RP} = \sum_{r \in R} \sum_{p \in P} W_{rp}, \tag{115}$$

with $r$ and $p$ nodes in the original graph $G$. We define the set of edges $\underline{\mathcal{E}}$ on $\underline{G}$ as

$$\underline{\mathcal{E}} = \{(R, P) \in \underline{\mathcal{G}} \times \underline{\mathcal{G}} : \underline{W}_{RP} > 0\} \tag{116}$$

and assign $\underline{W}_{RP}$ as weight to such edges. Node weights of limit nodes are defined similarly as aggregated weights of all nodes $r$ (in $G$) contained in the component $R$ as

$$\underline{\mu}_R = \sum_{r \in R} \mu_r. \tag{117}$$

In order to translate signals between the original graph $G$ and the limit description $\underline{G}$, we need translation operators mapping signals from one graph to the other:

**Definition E.2.** Denote by $\mathbb{1}_R$ the vector that has 1 as entries on nodes $r$ belonging to the connected (in $G_{\text{hign}}$) component $R$ and has entry zero for all nodes not in $R$. We define the down-projection operator $J^{\downarrow}$ component-wise via evaluating at node $R$ in $\underline{\mathcal{G}}$ as

$$(J^{\downarrow}x)_R = \langle \mathbb{1}_R, x \rangle / \underline{\mu}_R. \tag{118}$$

The upsampling operator $J^{\uparrow}$ is defined as

$$J^{\uparrow}u = \sum_R u_R \cdot \mathbb{1}_R; \tag{119}$$

where $u_R$ is a scalar value (the component entry of $u$ at $R \in \underline{\mathcal{G}}$) and the sum is taken over all connected components in $G_{\text{high}}$.

Slightly extending established results in Koke (2026), we then have the following result:

**Theorem E.3.** We have

$$\left\| R_z(L) - J^{\uparrow} R_z(\underline{L}) J^{\downarrow} \right\| = \mathcal{O}\left( \frac{\|L_{\text{reg.}}\|}{\lambda_1(L_{\text{high}})} \right) \tag{120}$$

holds; with $\lambda_1(L_{\text{high}})$ denoting the first non-zero eigenvalue of $L_{\text{high}}$.

$$\lambda_{\max}(L_{\text{reg.}}) = \|L_{\text{reg.}}\|. \tag{121}$$

We here inclue the proof of this fact for convenience and self-containedness.

*Proof.* We will split the proof of this result into multiple steps. For $z < 0$ Let us denote by

$$R_z(L) = (L - zId)^{-1}, \tag{122}$$
$$R_z(L_{high}) = (L_{high} - zId)^{-1} \tag{123}$$
$$R_z(L_{reg.}) = (L_{reg.} - zId)^{-1} \tag{124}$$

the resolvents correspodning to $L$, $L_{high}$ and $L_{reg.}$ respectively.
Our first goal is establishing that we may write

$$R_z(L) = [Id + R_z(L_{high})L_{reg.}]^{-1} \cdot R_z(L_{high}) \tag{125}$$

This will follow as a consequence of what is called the second resolvent formula Teschl (2014):

"Given self-adjoint operators $A, B$, we may write

$$R_z(A + B) - R_z(A) = -R_z(A)BR_z(A + B)." \tag{126}$$

In our case, this translates to

$$R_z(L) - R_z(L_{high}) = -R_z(L_{high})L_{reg.}R_z(L) \tag{127}$$

or equivalently

$$[Id + R_z(L_{high})L_{reg.}] R_z(L) = R_z(L_{high}). \tag{128}$$

Multiplying with $[Id + R_z(L_{high})L_{reg.}]^{-1}$ from the left then yields

$$R_z(L) = [Id + R_z(L_{high})L_{reg.}]^{-1} \cdot R_z(L_{high}) \tag{129}$$

as desired.
Hence we need to establish that $[Id + R_z(L_{high})L_{reg.}]$ is invertible for $z < 0$.

To establish a contradiction, assume it is not invertible. Then there is a signal $x$ such that

$$[Id + R_z(L_{high})L_{reg.}] x = 0. \tag{130}$$

Multiplying with $(L_{\text{high}} - zId)$ from the left yields

$$(L_{\text{high}} + L_{\text{reg.}} - zId)x = 0 \tag{131}$$

which is precisely to say that

$$(L - zId)x = 0 \tag{132}$$

But since $L$ is a graph Laplacian, it only has non-negative eigenvalues. Hence we have reached our contradiction and established

$$R_z(L) = [Id + R_z(L_{high})L_{reg.}]^{-1} R_z(L_{high}). \tag{133}$$

Our next step is to establish that

$$R_z(L_{high}) \to \frac{P_0^{\text{high}}}{-z}, \tag{134}$$

where $P_0^{\text{high}}$ is the spectral projection onto the eigenspace corresponding to the lowest lying eigenvalue $\lambda_0(L_{high}) = 0$ of $L_{high}$. Indeed, by the spectral theorem for finite dimensional operators (c.f. e.g. Teschl (2014)), we may write

$$R_z(\Delta_{high}) \equiv (L_{high} - zId)^{-1} = \sum_{\lambda \in \sigma(L_{high})} \frac{1}{\lambda - z} \cdot P_\lambda^{high}. \tag{135}$$

Here $\sigma(L_{high})$ denotes the spectrum (i.e. the collection of eigenvalues) of $L_{high}$ and the $\{P_\lambda^{high}\}_{\lambda \in \sigma(L_{high})}$ are the corresponding (orthogonal) eigenprojections onto the eigenspaces of the respective eigenvalues. Thus we find

$$\left\| R_z(L_{high}) - \frac{P_0^{high}}{-z} \right\| = \left\| \sum_{0 < \lambda \in \sigma(L_{high})} \frac{1}{\lambda - z} \cdot P_\lambda^{high} \right\|; \tag{136}$$

where the sum on the right hand side now excludes the eigenvalue $\lambda = 0$.

Using orthonormality of the spectral projections, the fact that $z < 0$ and monotonicity of $1/(\cdot + |z|)$ we find

$$\left\| R_z(L_{high}) - \frac{P_0^{high}}{-z} \right\| = \frac{1}{\lambda_1(\Delta_{high}) + |z|}. \tag{137}$$

Here $\lambda_1(L_{high})$ is the firt non-zero eigenvalue of $(L_{high})$.
Non-zero eigenvalues scale linearly with the weight scale since we have

$$\lambda(S \cdot L) = S \cdot \lambda(L) \tag{138}$$

for any graph Laplacian (in fact any matrix) $L$ with eigenvalue $\lambda$. Thus we have

$$\left\| R_z(L_{high}) - \frac{P_0^{high}}{-z} \right\| = \frac{1}{\lambda_1(L_{high}) + |z|} \leq \frac{1}{\lambda_1(L_{high})} \longrightarrow 0 \tag{139}$$

as $\lambda_1(L_{high}) \to \infty$.

Our next task is to use this result in order to bound the difference

$$I := \left\| \left[ Id + \frac{P_0^{high}}{-z} L_{reg.} \right]^{-1} \frac{P_0^{high}}{-z} - [Id + R_z(L_{high})L_{reg.}]^{-1} R_z(L_{high}) \right\|. \tag{140}$$

To this end we first note that the relation

$$[A + B - zId]^{-1} = [Id + R_z(A)B]^{-1} R_z(A) \tag{141}$$

provided to us by the second resolvent formula, implies

$$[Id + R_z(A)B]^{-1} = Id - B[A + B - zId]^{-1}. \tag{142}$$

Thus we have

$$\left\| [Id + R_z(L_{high})L_{reg.}]^{-1} \right\| \le 1 + \|L_{reg.}\| \cdot \|R_z(L)\| \tag{143}$$

$$\le 1 + \frac{\|L_{reg.}\|}{|z|}. \tag{144}$$

With this, we have

$$\left\| \left[ Id + \frac{P_0^{high}}{-z} L_{reg.} \right]^{-1} \cdot \frac{P_0^{high}}{-z} - R_z(L) \right\| \tag{145}$$

$$= \left\| \left[ Id + \frac{P_0^{high}}{-z} L_{reg.} \right]^{-1} \cdot \frac{P_0^{high}}{-z} - [Id + R_z(L_{high})L_{reg.}]^{-1} \cdot R_z(L_{high}) \right\| \tag{146}$$

$$\le \left\| \frac{P_0^{high}}{-z} \right\| \cdot \left\| \left[ Id + \frac{P_0^{high}}{-z} L_{reg.} \right]^{-1} - [Id + R_z(L_{high})L_{reg.}]^{-1} \right\| + \left\| \frac{P_0^{high}}{-z} - R_z(L_{high}) \right\| \cdot \left\| [Id + R_z(L_{high})L_{reg.}]^{-1} \right\| \tag{147}$$

$$\le \frac{1}{|z|} \left\| \left[ Id + \frac{P_0^{high}}{-z} L_{reg.} \right]^{-1} - [Id + R_z(L_{high})L_{reg.}]^{-1} \right\| + \left( 1 + \frac{\|L_{reg.}\|}{|z|} \right) \cdot \frac{1}{\lambda_1(L_{high})}. \tag{148}$$

Hence it remains to bound the left hand summand. For this we use the following fact (c.f. Horn & Johnson (2012), Section 5.8. "Condition numbers: inverses and linear systems"):

Given square matrices $A, B, C$ with $C = B - A$ and $\|A^{-1}C\| < 1$, we have

$$\|A^{-1} - B^{-1}\| \le \frac{\|A^{-1}\| \cdot \|A^{-1}C\|}{1 - \|A^{-1}C\|}. \tag{149}$$

In our case, this yields (together with $\|P_0^{high}\| = 1$) that

$$\left\| \left[ Id + P_0^{high}/(-z) \cdot L_{reg.} \right]^{-1} - [Id + R_z(L_{high})L_{reg.}]^{-1} \right\| \tag{150}$$

$$\le \frac{(1 + \|L_{reg.}\|/|z|)^2 \cdot \|L_{reg.}\| \cdot \|\frac{P_0^{high}}{-z} - R_z(L_{high})\|}{1 - (1 + \|L_{reg.}\|/|z|) \cdot \|L_{reg.}\| \cdot \|\frac{P_0^{high}}{-z} - R_z(L_{high})\|} \tag{151}$$

For $S_{high}$ sufficiently large, we have

$$\| - P_0^{high}/z - R_z(L_{high})\| \le \frac{1}{2(1 + \|L_{reg.}\|/|z|)} \tag{152}$$

so that we may estimate

$$\left\| \left[ Id + L_{reg.} \frac{P_0^{high}}{-z} \right]^{-1} - [Id + L_{reg.} R_z(L_{high})]^{-1} \right\| \tag{153}$$

$$\le 2 \cdot (1 + \|L_{reg.}\|) \cdot \|\frac{P_0^{high}}{-z} - R_z(L_{high})\| \tag{154}$$

$$= 2 \frac{1 + \|L_{reg.}\|/|z|}{\lambda_1(L_{high})} \tag{155}$$

Thus we have now established

$$\left\| \left[ Id + \frac{P_0^{high}}{-z} L_{reg.} \right]^{-1} \cdot \frac{P_0^{high}}{-z} - R_z(L) \right\| = \mathcal{O} \left( \frac{\|L_{reg.}\|}{\lambda_1(L_{high})} \right). \tag{156}$$

Hence we are done with the proof, as soon as we can establish

$$\left[ -zId + P_0^{high} L_{reg.} \right]^{-1} P_0^{high} = J^\uparrow R_z(\underline{L}) J^\downarrow, \tag{157}$$

with $J^\uparrow, \underline{L}, J^\downarrow$ as defined above. To this end, we first note that

$$J^\uparrow \cdot J^\downarrow = P_0^{high} \tag{158}$$

and

$$J^\downarrow \cdot J^\uparrow = Id_{\underline{G}}. \tag{159}$$

Indeed, the relation (158) follows from the fact that the eigenspace corresponding to the eignvalue zero is spanned by the vectors $\{\mathbb{1}_R\}_R$, with $\{R\}$ the connected components of $G_{\text{high}}$. Equation (159) follows from the fact that

$$\langle \mathbb{1}_R, \mathbb{1}_R \rangle = \underline{\mu}_R. \tag{160}$$

With this we have

$$\left[ Id + P_0^{high} L_{reg.} \right]^{-1} P_0^{high} = \left[ Id + J^\uparrow J^\downarrow L_{reg.} \right]^{-1} J^\uparrow J^\downarrow. \tag{161}$$

To proceed, set

$$\underline{x} := F^\downarrow x \tag{162}$$

and

$$\mathscr{X} = \left[ P_0^{high} L_{reg.} - zId \right]^{-1} P_0^{high} x. \tag{163}$$

Then

$$\left[ P_0^{high} L_{reg.} - zId \right] \mathscr{X} = P_0^{high} x \tag{164}$$

and hence $\mathscr{X} \in \text{Ran}(P_0^{high})$. Thus we have

$$J^\uparrow J^\downarrow (L_{\text{reg.}} - zId) J^\uparrow J^\downarrow \mathscr{X} = J^\uparrow J^\downarrow x. \tag{165}$$

Multiplying with $J^\downarrow$ from the left yields

$$J^\downarrow (L_{\text{reg.}} - zId) J^\uparrow J^\downarrow \mathscr{X} = J^\downarrow x. \tag{166}$$

Thus we have

$$(J^\downarrow L_{\text{reg.}} J^\uparrow - zId) J^\uparrow J^\downarrow \mathscr{X} = J^\downarrow x. \tag{167}$$

This – in turn – implies

$$J^\uparrow J^\downarrow \mathscr{X} = \left[ J^\downarrow L_{\text{reg.}} J^\uparrow - zId \right]^{-1} J^\downarrow x. \tag{168}$$

Using

$$P_0^{high} \mathscr{X} = \mathscr{X}, \tag{169}$$

we then have

$$\mathscr{X} = J^\uparrow \left[ J^\downarrow L_{\text{reg.}} J^\uparrow - zId \right]^{-1} J^\downarrow x. \tag{170}$$

We have thus concluded the proof if we can prove that $J^\downarrow L_{\text{reg.}} J^\uparrow$ is the Laplacian corresponding to the graph $\underline{G}$ defined in Definition E.1. But this is a straightforward calculation. $\square$

As a corollary, one finds

**Corollary E.4.** *We have*

$$R_z(L)^k \to J^\uparrow R^k(\underline{L}) J^\downarrow \tag{171}$$

*Proof.* This follows directly from the fact that

$$J^\downarrow J^\uparrow = Id_{\underline{G}}. \tag{172}$$

$\square$

