# OpenReview forum: "Scale Continuity in Graph Learning: Going beyond spectral methods"
_ICLR.cc/2026/Workshop/GRaM — ICLR 2026 Workshop GRaM Poster_

### Official Review · Reviewer_Z9YL · 2026-02-18
**Review for "Scale Continuity in Graph Learning: Going beyond spectral methods"**

**Rating:** 6
**Confidence:** 3

**Review:**

**Summary:**

The paper argues that scale continuity is not exclusive to spectral GNNs. It shows standard adjacency-based MPNNs can be discontinuous across scales. They propose a modified message passing framework that replaces adjacency-based propagation with a global Laplacian propagation operator derived from Laplace-transform / heat-kernel style operators.


**Strengths:**

* Clear motivation and a concrete failure mode for standard MPNNs under coarsening/refinement.
* Simple and elegant idea: continuity depends on the propagation operator, not "spectral vs message passing".
* Empirical evidence is aligned with theoretical analysis.

**Weaknesses**

* Practical computation/approximation of $\psi(L)$ (e.g., $e^{-tL}$) and sparsification/runtime are not fully quantified.
* Experiments are narrow (mainly QM7 with a specific coarsening scheme); limited robustness evidence.

**Pmlr Suitability:**

NA

---

### Official Review · Reviewer_WNDT · 2026-02-23
**A nice extension of recent ideas to message passing**

**Rating:** 7
**Confidence:** 3

**Review:**

The authors consider the problem of scale continuity in message passing neural networks. This refers to networks that react gracefully to using graphs representing the same underlying object, but changing the resolutions. The authors use the illustrative example of graphs modelling molecules with and without clusters of atoms being merged into single nodes in the graph. If a message passing neural network is used naively for such a case, the network ends up not being continuous. The authors showcase this nicely with a numerical experiment on the QM9-dataset.

Recently, spectral methods have been proposed as a way to mitigate this problem, where the main idea is to utilize that the heat kernels of the graph react continuously to merging procedures as the one above. In this paper, the authors realize that this idea also can be used to make scale-continuous message passing networks -- if the aggregation procedure is done with weights obtained from functions depending on the heat kernel (specifically, using Laplace transforms of the Laplace matrix of the graph). They both provide a theoretical analysis of the procedure as well as a convincing numerical one.

I think this article is nicely written, with a clear message. Papers like this are important -- mantras like 'message passing cannot do scale continuity' may easily hinder progress, and a paper like this mitigates that.

As for weaknesses, the paper, in particular the appendix, would benifit from one more round of proofreading. Also, the paper could be deepened by discussing more explicitly what conditions need to be fulfilled in order for the difference in the heat kernels to vanish in the 'scale changing limits'.

**Pmlr Suitability:**

NA

---

### Meta-Review · Area_Chair_Zs74 · 2026-02-27

**Decision:**

Accept

**Metareview:**

This is a solid paper for the tiny paper track.

**Relevance To Proceedings:**

Tiny paper — does not apply

**Relevance To Workshop:**

Yes — suitable for GRaM

---

### Decision · Program_Chairs · 2026-03-02

Accept (Poster)